# Cyclic Counterfactuals under Shift–Scale Interventions

**Saptarshi Saha** [*]
Computer Vision and Pattern Recognition Unit
Indian Statistical Institute
Kolkata, West Bengal - 700108, India
`saptarshi.saha_r@isical.ac.in`

**Dhruv Vansraj Rathore**
Indian Statistical Institute
Kolkata, West Bengal - 700108, India
`cs2306@isical.ac.in`

**Utpal Garain**
Computer Vision and Pattern Recognition Unit
Indian Statistical Institute
Kolkata, West Bengal - 700108, India
`utpal@isical.ac.in`

## Abstract

Most counterfactual inference frameworks traditionally assume acyclic structural causal models (SCMs), i.e. directed acyclic graphs (DAGs). However, many real-world systems (e.g. biological systems) contain feedback loops or cyclic dependencies that violate acyclicity. In this work, we study counterfactual inference in cyclic SCMs under *shift–scale interventions*, i.e., soft, policy-style changes that rescale and/or shift a variable's mechanism.

## 1  Introduction

Most research on counterfactual reasoning (Pawlowski et al., 2020; Sanchez and Tsaftaris, 2022; Saha and Garain, 2022; Komanduri et al., 2024; Melistas et al., 2024; Wu et al., 2025; Kügelgen et al., 2023; Kladny et al., 2024) assumes that the underlying causal structure among variables can be represented by a Directed Acyclic Graph (DAG). However, this acyclicity assumption is often violated in real-world systems. For instance, gene regulatory networks frequently exhibit feedback loops, leading to cyclic dependencies that DAGs cannot capture. Such cycles are integral to the dynamic behavior of biological systems and are crucial for understanding processes like cell differentiation and immune responses. Given the prevalence of cycles in such systems and the availability of detailed perturbation data, there is a compelling case for extending counterfactual inference frameworks to accommodate cyclic structures.

However, progress in this area has stalled due to the need for new theoretical breakthroughs, as many properties that hold in acyclic models no longer apply when feedback loops are present. A fundamental issue is that a set of structural equations with cycles may not have a unique solution for the endogenous variables. Solvability refers to the existence of at least one solution (equilibrium), and unique solvability means there is exactly one solution (almost surely). If an SCM is not uniquely solvable —i.e., not a simple SCM—it might generate multiple different outcome distributions or undefined behavior under interventions. Although the class of simple SCMs includes acyclic SCMs as a special case, the theory remains much less developed for the cyclic setting.

In causality, a hard (structural) intervention (Pearl's do-operator) (Pearl et al., 2016) sets a variable to a fixed value, severing its dependence on its usual causes. In contrast, shift and scale interventions

---

[*]first author

are types of soft (parametric) interventions that modify the value of a variable by some function (such as adding or multiplying by a constant) without removing its original input links. In particular, a hard intervention is just a degenerate case of a soft intervention (Massidda et al., 2023). This generality means one can ask nuanced "what-if" questions: What if everyone received $20\%$ more of the drug? What if we lowered each student's class size by 5? Such policies cannot be represented as a simple $do(X = x)$ since they depend on individuals' original $X$. Soft interventions are strictly more expressive in defining counterfactual worlds. More specifically, a shift can implement dynamic-like policies ("increase dose for those who had high risk") that static $do$-interventions cannot capture. Shift interventions have been used to learn causal cyclic graphs (Rothenhäusler et al., 2015), to match a desired causal state (Zhang et al., 2021). Lorch et al. (2024) use shift-scale intervention in causal modeling with stationary diffusions. However, the theoretical foundations supporting their use remain underdeveloped.

**Contributions.** In this work, we develop a theoretical framework for counterfactual inference in *cyclic causal models* under *shift–scale interventions*. Our main contributions are as follows:

- We show that *SCMs satisfying a global contraction condition* are *simple*—i.e., uniquely solvable with respect to every subset of variables—even in the presence of cycles.

- We prove that under *shift–scale interventions with bounded scale coefficients* (i.e., $|a_j| \leq 1$), the intervened *twin SCM remains uniquely solvable*, ensuring the well-posedness of counterfactual queries.

- We establish that this class of shift–scale interventions is *closed under composition*, making it algebraically stable for sequential interventional analysis.

- Under an additional *Lipschitz regularity condition in the exogenous noise*, we derive *sub-Gaussian tail bounds* for counterfactual functionals, showing that the distribution of counterfactual outcomes concentrates sharply around their mean.

## 2 Background and Problem Setup

This section provides a brief overview of SCMs and establishes the notational framework adopted throughout the paper. Our exposition aligns with the formalism presented in Bongers et al. (2021).

**Definition 1** (Structural Causal Model). *A Structural Causal Model (SCM) is defined as a tuple*

$$\mathcal{M} = \langle \mathcal{I}, \mathcal{J}, \mathcal{X}, \mathcal{E}, f, \mathbb{P}_{\mathcal{E}} \rangle,$$

*where $\mathcal{I}$ denotes a finite set indexing the endogenous variables, $\mathcal{J}$ denotes a disjoint finite set indexing the exogenous variables, $\mathcal{X} = \prod_{i \in \mathcal{I}} \mathcal{X}_i$ is the joint domain of the endogenous variables, with each $\mathcal{X}_i$ being a standard measurable space, $\mathcal{E} = \prod_{j \in \mathcal{J}} \mathcal{E}_j$ is the joint domain of the exogenous variables, with each $\mathcal{E}_j$ also a standard measurable space, $f : \mathcal{X} \times \mathcal{E} \to \mathcal{X}$ is a measurable function representing the causal mechanisms that determine the values of endogenous variables from both endogenous and exogenous inputs, $\mathbb{P}_{\mathcal{E}} = \prod_{j \in \mathcal{J}} \mathbb{P}_{\mathcal{E}_j}$ is a product probability measure over $\mathcal{E}$, describing the joint distribution of the exogenous variables.*

**Definition 2** (Solution of an SCM). *A solution of $\mathcal{M}$ is a pair of random variables $(\mathbf{X}, \mathbf{E})$ defined on a common probability space $(\Omega, \mathcal{F}, \mathbb{P})$ such that:*

*(i)* $\mathbf{E} : \Omega \to \mathcal{E}$ *has distribution* $\mathbb{P}_{\mathcal{E}}$;

*(ii)* $\mathbf{X} : \Omega \to \mathcal{X}$ *satisfies the structural equations*

$$\mathbf{X} = f(\mathbf{X}, \mathbf{E}) \quad \mathbb{P}\text{-almost surely.}$$

*For convenience, we say that a random variable $X$ is a solution of $\mathcal{M}$ if there exists an exogenous random variable $\mathbf{E}$ such that the pair $(\mathbf{X}, \mathbf{E})$ constitutes a solution of $\mathcal{M}$*

**Definition 3** (Parent). *For $i \in \mathcal{I}$, an index $k \in \mathcal{I} \cup \mathcal{J}$ is called a parent of $i$ iff there does not exist a measurable map $\tilde{f}_i : \mathcal{X}_{\setminus k} \times \mathcal{E}_{\setminus k} \to \mathcal{X}_i$ such that for $P_{\mathcal{E}}$-almost every $e \in \mathcal{E}$ and all $x \in \mathcal{X}$,*

$$x_i = f_i(x, e) \iff x_i = \tilde{f}_i(x_{\setminus k}, e_{\setminus k}).$$

*Exogenous variables have no parents. We write $\mathrm{pa}(i)$ for the set of parents of $i$ and extend to sets by $\mathrm{pa}(O) := \bigcup_{i \in O} \mathrm{pa}(i)$.*

**Definition 4** (Unique Solvability)**.** *An SCM $\mathcal{M}$ is* uniquely solvable with respect to *a subset $\mathcal{O} \subseteq \mathcal{I}$ if there exists a measurable function*

$$g_{\mathcal{O}} : \mathcal{X}_{\mathrm{pa}(\mathcal{O}) \setminus \mathcal{O}} \times \mathcal{E}_{\mathrm{pa}(\mathcal{O})} \to \mathcal{X}_{\mathcal{O}}$$

*such that for all $x \in \mathcal{X}$ and $\mathbb{P}_{\mathcal{E}}$-almost every $e \in \mathcal{E}$,*

$$x_{\mathcal{O}} = g_{\mathcal{O}}(x_{\mathrm{pa}(\mathcal{O}) \setminus \mathcal{O}}, e_{\mathrm{pa}(\mathcal{O})}) \quad \Longleftrightarrow \quad x_{\mathcal{O}} = f_{\mathcal{O}}(x, e).$$

**Definition 5** (Simple SCM)**.** *An SCM $\mathcal{M}$ is called* simple *if it is uniquely solvable with respect to every subset $\mathcal{O} \subseteq \mathcal{I}$.*

Acyclic SCMs are simple.

**Definition 6** (Twin SCM)**.** *Let $\mathcal{M}$ be a structural causal model. The* twin SCM *associated with $\mathcal{M}$ is defined as*

$$\mathcal{M}^{\mathrm{twin}} := \langle \mathcal{I} \cup \mathcal{I}', \ \mathcal{J}, \ \mathcal{X} \times \mathcal{X}, \ \mathcal{E}, \ \tilde{f}, \ \mathbb{P}_{\mathcal{E}} \rangle,$$

*where $\mathcal{I}' := \{ i' : i \in \mathcal{I} \}$ is a disjoint copy of the endogenous index set, and $\tilde{f} : \mathcal{X} \times \mathcal{X} \times \mathcal{E} \to \mathcal{X} \times \mathcal{X}$ is the measurable function defined by*

$$\tilde{f}(x, x', e) := \big( f(x, e), \ f(x', e) \big),$$

*with $x, x' \in \mathcal{X}$ and $e \in \mathcal{E}$.*

**How the twin map $\tilde{f}$ is constructed** For any noise realisation $e \in \mathcal{E}$ and stacked endogenous state $(x, x') = \big( x_1, \ldots, x_{|\mathcal{I}|}, \ x'_1, \ldots, x'_{|\mathcal{I}|} \big) \in \mathcal{X} \times \mathcal{X}$, the twin-SCM mechanism $\tilde{f} : \mathcal{X} \times \mathcal{X} \times \mathcal{E} \longrightarrow \mathcal{X} \times \mathcal{X}$ is defined by $\tilde{f}(x, x', e) := \big( f(x, e), \ f(x', e) \big)$. Written coordinate-wise:

$$\tilde{f}_j(x, x', e) = \begin{cases} f_j(x, e), & j \in \mathcal{I}, \\ f_i(x', e), & j = i' \text{ for some } i \in \mathcal{I} (\text{i.e., } j \in \mathcal{I}'), \end{cases}$$

i.e. the first copy (un-primed) follows the original mechanism $X_j \leftarrow f_j(x, e)$, while the primed copy applies the same function $f_i$ to its own state $x'$. In compact notation

$$\tilde{f}_j(x, x', e) = \begin{cases} f_j(x, e), & j \in \mathcal{I}, \\ f_i(x', e), & j' \in \mathcal{I}'. \end{cases}$$

**Definition 7** (Counterfactual distribution )**.** *Let $\mathcal{M} = \langle \mathcal{I}, \mathcal{J}, \mathcal{X}, \mathcal{E}, f, \mathbb{P}_{\mathcal{E}} \rangle$ be an SCM and let $\mathcal{M}^{\mathrm{twin}}$ be its twin SCM. Consider a perfect intervention*

$$\mathrm{do}\big( \tilde{\mathcal{I}}, \ \xi_{\tilde{\mathcal{I}}} \big), \qquad \tilde{\mathcal{I}} \subseteq \mathcal{I} \cup \mathcal{I}', \quad \xi_{\tilde{\mathcal{I}}} \in \mathcal{X}_{\tilde{\mathcal{I}}},$$

*applied to $\mathcal{M}^{\mathrm{twin}}$, and denote the intervened model by $\big( \mathcal{M}^{\mathrm{twin}} \big)_{\mathrm{do}(\tilde{\mathcal{I}}, \xi_{\tilde{\mathcal{I}}})}$. If this intervened twin SCM admits a (measurable) solution $(X, X') \in \mathcal{X} \times \mathcal{X}$, then the joint distribution $\mathbb{P}_{(X, X')}$ is called the* counterfactual distribution *of $\mathcal{M}$ under the perfect intervention $\mathrm{do}(\tilde{\mathcal{I}}, \xi_{\tilde{\mathcal{I}}})$ associated with the pair of random variables $(X, X')$.*

In the appendix, we delineate a clear correspondence between the twin-network formulation of structural causal models and Pearl's canonical action–abduction–prediction schema for counterfactual inference.

## 2.1 Shift–Scale intervention

For example, instead of forcing a treatment $X$ to a set value, a shift intervention might increase each individual's natural treatment dose by a fixed amount $\delta$, and a scale intervention might multiply each dose by a factor (e.g. $10\%$ increase) – all while allowing $X$ to remain influenced by its usual causes (confounders, prior variables, etc.). This preserves the causal edges into $X$ but changes the conditional distribution or structural equation of $X$.

**Definition 8** (Shift–Scale intervention). *Let $\mathcal{M} = \langle \mathcal{I}, \mathcal{J}, \mathcal{X}, \mathcal{E}, f, \mathbb{P}_{\mathcal{E}} \rangle$ be an SCM and fix a non-empty subset $\tilde{\mathcal{I}} \subseteq \mathcal{I}$. For each $j \in \tilde{\mathcal{I}}$ choose scale $a_j \in \mathbb{R}$ and shift $b_j \in \mathbb{R}$. The shift–scale intervention*

$$\mathrm{ss}\big(\tilde{\mathcal{I}}, a_{\tilde{\mathcal{I}}}, b_{\tilde{\mathcal{I}}}\big) \quad \big(a_{\tilde{\mathcal{I}}} := (a_j)_{j \in \tilde{\mathcal{I}}}, \ b_{\tilde{\mathcal{I}}} := (b_j)_{j \in \tilde{\mathcal{I}}}\big)$$

*produces a new SCM*

$$\mathcal{M}_{\mathrm{ss}} := \langle \mathcal{I}, \mathcal{J}, \mathcal{X}, \mathcal{E}, f^{\mathrm{ss}}, \mathbb{P}_{\mathcal{E}} \rangle, \qquad f_i^{\mathrm{ss}}(x, e) := \begin{cases} a_i\, f_i(x, e) + b_i, & i \in \tilde{\mathcal{I}}, \\ f_i(x, e), & i \notin \tilde{\mathcal{I}}. \end{cases}$$

Perfect or $\mathrm{do}$-intervention is recovered as the special case $a_j = 0, \ b_j = \xi_j$.

**Definition 9** (Shift–Scale counterfactual distribution). *Let $\mathcal{M}^{\mathrm{twin}}$ be the twin SCM of $\mathcal{M}$. Apply the shift–scale intervention only to the first copy:*

$$\big(\mathcal{M}^{\mathrm{twin}}\big)_{\mathrm{ss}} := \big(\mathcal{M}^{\mathrm{twin}}\big)_{\mathrm{ss}\big(\tilde{\mathcal{I}}, a_{\tilde{\mathcal{I}}}, b_{\tilde{\mathcal{I}}}\big)}.$$

*If this intervened twin SCM admits a measurable solution $(X, X') \in \mathcal{X} \times \mathcal{X}$, we call the joint law $\mathbb{P}_{(X, X')}$ the* shift–scale counterfactual distribution *of $\mathcal{M}$ under the intervention $\mathrm{ss}\big(\tilde{\mathcal{I}}, a_{\tilde{\mathcal{I}}}, b_{\tilde{\mathcal{I}}}\big)$.*

Given a subset a set of coordinates $\tilde{\mathcal{I}} \subseteq \mathcal{I} \cup \mathcal{I}'$, and parameters $a_{\tilde{\mathcal{I}}}, b_{\tilde{\mathcal{I}}} \in \mathbb{R}^{|\tilde{\mathcal{I}}|}$, we want to replace the structural equation

$$X_j = \tilde{f}_j(\cdot) \quad \longrightarrow \quad X_j = a_j\, \tilde{f}_j(\cdot) + b_j,$$

for each $j \in \tilde{\mathcal{I}}$. All other coordinates stay unchanged. Encoding these modifications into a single map

$$\tilde{g}(x, x', e) := \big(g_j(x, x', e)\big)_{j \in \mathcal{I} \cup \mathcal{I}'}, \qquad g_j(x, x', e) := \begin{cases} a_j\, \tilde{f}_j(x, x', e) + b_j, & j \in \tilde{\mathcal{I}}, \\ \tilde{f}_j(x, x', e), & j \notin \tilde{\mathcal{I}}, \end{cases}$$

gives the intervened twin update rule. Specifically

$$\text{if } j \in \mathcal{I} \ \Rightarrow \ g_j(x, x', e) = a_j\, f_j(x, e) + b_j,$$
$$\text{if } j = i' \in \mathcal{I}' \ \Rightarrow \ g_{i'}(x, x', e) = a_{i'}\, f_i(x', e) + b_{i'}.$$

Thus each copy (un-primed and primed) is modified *only* on the requested coordinates, allowing independent interventions on the two worlds.

## 2.2 Semantics of Cyclic SCMs

One natural interpretation of cyclic structural equations is by assuming an underlying discrete-time dynamical system, where the equations act as update rules: the value of each variable at time $t{+}1$ is computed from the values at time $t$. The system is then analyzed in the limit as $t \to \infty$, focusing on the fixed points to which the dynamics converge. Mooij et al. (2013) demonstrate that an alternative, yet natural, interpretation of SCMs emerges when considering systems of ordinary differential equations (ODEs). By examining the equilibrium (steady-state) solutions of such ODEs, one arrives at a structural causal model that is time-independent, but still retains meaningful causal semantics with respect to interventions. Specifically, the semantics of interventions and counterfactuals (see also Appendix A.1) remain valid and well-defined in this steady-state context, as rigorously formalized by Mooij et al. (2013) and further extended by Bongers et al. (2021). These are by no means the only routes to structural causal models; indeed, SCMs may arise through a variety of alternative constructions and representations, depending on the nature of the system under consideration. Although many physical processes exhibit inertia, static cyclic structural causal models (SCMs) remain appropriate when we focus on equilibrium behavior or sample at a temporal resolution coarser than the fastest feedback loop. Examples include gene-regulatory networks in single-cell genomics (Rohbeck et al., 2024); market-equilibrium models (Bongers et al., 2021); predator–prey ecological systems; Thyroid or reproductive hormone axes exhibit feedback loops (Clarke et al., 2014), etc. Our mathematical framework is agnostic about the interpretation of cycles: we formulate everything directly at the level of structural equations with exogenous noise.

# 3 Theory

**Theorem 1** (Global $\ell^p$-contraction $\implies$ simple SCM). *Let $\mathcal{M} = \langle \mathcal{I}, \mathcal{J}, \mathcal{X}, \mathcal{E}, f, \mathbb{P}_{\mathcal{E}} \rangle$ be an SCM whose endogenous index set $\mathcal{I}$ is finite. Assume each coordinate domain ($\mathcal{X}_i \subseteq \mathbb{R}$) is non-empty and* closed. *Fix $p \in [1, \infty]$ and endow every product space with the $\ell^p$-norm $\|x\|_p := (\sum_{i \in \mathcal{I}} |x_i|^p)^{1/p}$ (for $p = \infty$ take the maximum norm). Suppose there exists $\kappa \in [0, 1)$ such that*

$$\|f(x, e) - f(y, e)\|_p \leq \kappa \|x - y\|_p \quad \text{for all } x, y \in \mathcal{X}, \ e \in \mathcal{E}. \tag{1}$$

*Then $\mathcal{M}$ is uniquely solvable with respect to* every *subset $\mathcal{O} \subseteq \mathcal{I}$, and hence $\mathcal{M}$ is a* simple *SCM.*

*Proof.* For any subset $\mathcal{O} \subseteq \mathcal{I}$ let $\mathcal{Q} := \mathrm{pa}(\mathcal{O}) \setminus \mathcal{O}$. Because every $\mathcal{X}_i$ is closed in $\mathbb{R}$ (hence complete) and $\mathcal{O}$ is finite, the product $\mathcal{X}_{\mathcal{O}} = \prod_{i \in \mathcal{O}} \mathcal{X}_i$ is complete under the $\ell^p$-metric; the same holds for the full space $\mathcal{X}$.

For each pair $(x_{\mathcal{Q}}, e_{\mathrm{pa}(\mathcal{O})})$ define

$$h_{x_{\mathcal{Q}}, e_{\mathrm{pa}(\mathcal{O})}} : \mathcal{X}_{\mathcal{O}} \longrightarrow \mathcal{X}_{\mathcal{O}}, \qquad u \mapsto f_{\mathcal{O}}\big(u, x_{\mathcal{Q}}, x_{\mathcal{I} \setminus (\mathcal{O} \cup \mathcal{Q})}, e_{\mathrm{pa}(\mathcal{O})}, e_{\mathcal{J} \setminus \mathrm{pa}(\mathcal{O})}\big),$$

where the "dummy" coordinates $x_{\mathcal{I} \setminus (\mathcal{O} \cup \mathcal{Q})}$ and $e_{\mathcal{J} \setminus \mathrm{pa}(\mathcal{O})}$ may be chosen arbitrarily because they do not influence $f_{\mathcal{O}}$.

For $u, v \in \mathcal{X}_{\mathcal{O}}$ define $\tilde{u} := (u, x_{\mathcal{Q}}, x_{\mathcal{I} \setminus (\mathcal{O} \cup \mathcal{Q})})$, $\tilde{v} := (v, x_{\mathcal{Q}}, x_{\mathcal{I} \setminus (\mathcal{O} \cup \mathcal{Q})}) \in \mathcal{X}$. Then $\|\tilde{u} - \tilde{v}\|_p = \|u - v\|_p$ because $\tilde{u}, \tilde{v}$ coincide outside $\mathcal{O}$. Using (1),

$$\|h_{x_{\mathcal{Q}}, e_{\mathrm{pa}(\mathcal{O})}}(u) - h_{x_{\mathcal{Q}}, e_{\mathrm{pa}(\mathcal{O})}}(v)\|_p = \|f_{\mathcal{O}}(\tilde{u}, e) - f_{\mathcal{O}}(\tilde{v}, e)\|_p \leq \|f(\tilde{u}, e) - f(\tilde{v}, e)\|_p \leq \kappa \|u - v\|_p.$$

Thus each map $h_{x_{\mathcal{Q}}, e_{\mathrm{pa}(\mathcal{O})}}$ is a $\kappa$-contraction on the *complete* metric space $(\mathcal{X}_{\mathcal{O}}, \|\cdot\|_p)$. By the Banach fixed-point theorem, for every $(x_{\mathcal{Q}}, e_{\mathrm{pa}(\mathcal{O})})$ there exists a *unique* element $u^*(x_{\mathcal{Q}}, e_{\mathrm{pa}(\mathcal{O})}) \in \mathcal{X}_{\mathcal{O}}$ satisfying $u^* = h_{x_{\mathcal{Q}}, e_{\mathrm{pa}(\mathcal{O})}}(u^*)$.

**Measurability.** Fix any $\bar{u} \in \mathcal{X}_{\mathcal{O}}$. Defining Picard iterates $u^{(0)} \equiv \bar{u}$ and $u^{(n+1)} := h_{x_{\mathcal{Q}}, e_{\mathrm{pa}(\mathcal{O})}}(u^{(n)})$, one obtains a sequence of measurable functions converging *pointwise* to $u^*$. Limits of measurable functions are measurable, hence the map

$$g_{\mathcal{O}}(x_{\mathcal{Q}}, e_{\mathrm{pa}(\mathcal{O})}) := u^*(x_{\mathcal{Q}}, e_{\mathrm{pa}(\mathcal{O})})$$

is measurable.

**Equivalence.** ($\Rightarrow$) By definition, $g_{\mathcal{O}}(x_{\mathcal{Q}}, e_{\mathrm{pa}(\mathcal{O})})$ is the unique fixed point $u^*$ of the map

$$h_{x_{\mathcal{Q}}, e_{\mathrm{pa}(\mathcal{O})}}(u) := f_{\mathcal{O}}(u, x_{\mathcal{Q}}, x_{\mathcal{I} \setminus (\mathcal{O} \cup \mathcal{Q})}, e_{\mathrm{pa}(\mathcal{O})}, e_{\mathcal{J} \setminus \mathrm{pa}(\mathcal{O})}).$$

Hence, if $x_{\mathcal{O}} = g_{\mathcal{O}}(x_{\mathcal{Q}}, e_{\mathrm{pa}(\mathcal{O})})$, then $x_{\mathcal{O}} = h_{x_{\mathcal{Q}}, e_{\mathrm{pa}(\mathcal{O})}}(x_{\mathcal{O}}) = f_{\mathcal{O}}(x, e)$.

($\Leftarrow$) Suppose $x_{\mathcal{O}} = f_{\mathcal{O}}(x, e)$. Then $x_{\mathcal{O}}$ satisfies $x_{\mathcal{O}} = f_{\mathcal{O}}(x, e) = h_{x_{\mathcal{Q}}, e_{\mathrm{pa}(\mathcal{O})}}(x_{\mathcal{O}})$, so it is a fixed point of $h_{x_{\mathcal{Q}}, e_{\mathrm{pa}(\mathcal{O})}}$. By Banach's theorem, the fixed point is unique, so

$$x_{\mathcal{O}} = u^* = g_{\mathcal{O}}(x_{\mathcal{Q}}, e_{\mathrm{pa}(\mathcal{O})}).$$

Therefore,

$$x_{\mathcal{O}} = g_{\mathcal{O}}(x_{\mathcal{Q}}, e_{\mathrm{pa}(\mathcal{O})}) \quad \Longleftrightarrow \quad x_{\mathcal{O}} = f_{\mathcal{O}}(x, e),$$

establishing the defining equivalence in the notion of unique solvability holds for all $x$ and for $\mathbb{P}_{\mathcal{E}}$-almost every $e$.

Because $\mathcal{O} \subseteq \mathcal{I}$ was arbitrary, the same reasoning applies to every subset, establishing that $\mathcal{M}$ is uniquely solvable with respect to all subsets and hence is a simple SCM. $\square$

The global contraction condition implies unique solvability for every subset; hence the models are simple SCMs in the sense of Bongers et al. (2021). This places us directly inside the setting where their closure results for do-interventions, marginalization and twin networks apply.

---

**Theorem 2** (Unique Solvability of Twin SCM under Shift–Scale Interventions). *Let $\mathcal{M} = \langle \mathcal{I}, \mathcal{J}, \mathcal{X}, \mathcal{E}, f, \mathbb{P}_{\mathcal{E}} \rangle$ be an SCM such that the causal mechanism $f : \mathcal{X} \times \mathcal{E} \to \mathcal{X}$ satisfies a global $\ell^p$-contraction for some constant $0 \leq \kappa < 1$. Let $\mathcal{M}^{\text{twin}}$ be the associated twin SCM, and fix a subset $\tilde{\mathcal{I}} \subseteq \mathcal{I} \cup \mathcal{I}'$, along with shift–scale coefficients $a_{\tilde{\mathcal{I}}} \in \mathbb{R}^{|\tilde{\mathcal{I}}|}$, $b_{\tilde{\mathcal{I}}} \in \mathbb{R}^{|\tilde{\mathcal{I}}|}$, satisfying:*

$$a_{\max} := \sup_{j \in \tilde{\mathcal{I}}} |a_j| \leq 1.$$

*Then the shift–scale intervened twin SCM $\left( \mathcal{M}^{\text{twin}} \right)_{\text{ss}(\tilde{\mathcal{I}}, a_{\tilde{\mathcal{I}}}, b_{\tilde{\mathcal{I}}})}$ is uniquely solvable with respect to every subset of endogenous variables, and is thus a simple SCM. In particular, the associated counterfactual distribution $\mathbb{P}_{(X, X')}$ is well-defined.*

---

*Proof.* Let $(x, x') \in \mathcal{X} \times \mathcal{X}$ be the endogenous variables of the twin SCM and the twin map $\tilde{f}(x, x', e) := (f(x, e), f(x', e))$. is a global $\kappa$-contraction on $\mathcal{X} \times \mathcal{X}$:

$$\|\tilde{f}(x, x', e) - \tilde{f}(y, y', e)\|_p = \left( \|f(x, e) - f(y, e)\|_p^p + \|f(x', e) - f(y', e)\|_p^p \right)^{1/p} \leq \kappa \|(x, x') - (y, y')\|_p.$$

Now define the shift–scale intervened map $\tilde{g}(x, x', e)$ by:

$$\tilde{g}_j(x, x', e) := \begin{cases} a_j \tilde{f}_j(x, x', e) + b_j, & j \in \tilde{\mathcal{I}}, \\ \tilde{f}_j(x, x', e), & j \notin \tilde{\mathcal{I}}. \end{cases}$$

We now show that $\tilde{g}(\cdot, \cdot, e)$ is a global contraction in the $\ell^p$-norm with the same constant $\kappa$. Let $u := (x, x')$, $v := (y, y')$. Then for any $j \in \mathcal{I} \cup \mathcal{I}'$,

$$|\tilde{g}_j(u, e) - \tilde{g}_j(v, e)| = \begin{cases} |a_j| \cdot |\tilde{f}_j(u, e) - \tilde{f}_j(v, e)| \leq a_{\max} |\tilde{f}_j(u, e) - \tilde{f}_j(v, e)|, & j \in \tilde{\mathcal{I}}, \\ |\tilde{f}_j(u, e) - \tilde{f}_j(v, e)|, & j \notin \tilde{\mathcal{I}}. \end{cases}$$

Therefore,

$$\|\tilde{g}(u, e) - \tilde{g}(v, e)\|_p \leq \left( \sum_{j \in \tilde{\mathcal{I}}} (a_{\max} |\tilde{f}_j(u, e) - \tilde{f}_j(v, e)|)^p + \sum_{j \notin \tilde{\mathcal{I}}} |\tilde{f}_j(u, e) - \tilde{f}_j(v, e)|^p \right)^{1/p}.$$

Since $a_{\max} \leq 1$, we have:

$$\|\tilde{g}(u, e) - \tilde{g}(v, e)\|_p \leq \|\tilde{f}(u, e) - \tilde{f}(v, e)\|_p \leq \kappa \|u - v\|_p.$$

Thus $\tilde{g}(\cdot, e)$ is also a global $\kappa$-contraction on $(\mathcal{X} \times \mathcal{X}, \| \cdot \|_p)$, a complete metric space since $\mathcal{X}$ is closed and finite-dimensional.

By Banach's fixed-point theorem (as used in Theorem 1), the shift–scale intervened twin SCM has a unique solution $(X, X')$ for each $e$, and the solution map is measurable.

Since the SCM is uniquely solvable with respect to all subsets of $\mathcal{I} \cup \mathcal{I}'$, the intervened twin SCM is simple. Pushing forward $\mathbb{P}_{\mathcal{E}}$ through this solution map defines the shift–scale counterfactual distribution $\mathbb{P}_{(X, X')}$. $\qquad \square$

---

Proposition 8.2 in Bongers et al. (2021) shows that the class of simple SCMs is closed under (i) marginalization, (ii) perfect interventions (do), and (iii) the twin construction. It does not give a sufficient condition for simplicity; it presupposes simplicity. Theorem 2 gives sufficient analytic conditions for unique solvability of the twin under a class of soft, parametric shift–scale interventions, thereby ensuring well-posed counterfactuals beyond hard do.

**Proposition 1** (Closure under (composed) Shift–Scale Interventions)**.** *Let* $\mathcal{M} = \langle \mathcal{I}, \mathcal{J}, \mathcal{X}, \mathcal{E}, f, \mathbb{P}_{\mathcal{E}} \rangle$ *be an SCM such that for some* $p \in [1, \infty]$

$$\|f(x, e) - f(y, e)\|_p \leq \kappa \|x - y\|_p, \qquad \forall x, y \in \mathcal{X}, \ \forall e \in \mathcal{E},$$

*with a constant* $\kappa \in [0, 1)$*. Consider a* finite sequence *of shift–scale interventions applied successively to* $\mathcal{M}$*:*

$$\mathrm{ss}\big(\tilde{\mathcal{I}}^{(1)}, a^{(1)}, b^{(1)}\big) \circ \ldots \circ \mathrm{ss}\big(\tilde{\mathcal{I}}^{(m)}, a^{(m)}, b^{(m)}\big), \qquad m \geq 1,$$

*where for every* $r = 1, \ldots, m$ *and every* $j \in \tilde{\mathcal{I}}^{(r)}$ *one has* $|a_j^{(r)}| \leq 1$*.*

*Then the resulting (composed) intervened model is equivalent to a single shift–scale intervention* $\mathrm{ss}\big(\tilde{\mathcal{I}}, a^{\mathrm{comp}}, b^{\mathrm{comp}}\big)$ *with* $|a_j^{\mathrm{comp}}| \leq 1$ *for all* $j \in \tilde{\mathcal{I}}$*, and its structural function* $g(x, e) = a^{\mathrm{comp}} \odot f(x, e) + b^{\mathrm{comp}}$ *satisfies*

$$\|g(x, e) - g(y, e)\|_p \leq \kappa \|x - y\|_p, \qquad \forall x, y \in \mathcal{X}, \ e \in \mathcal{E}.$$

*Consequently the intervened SCM remains* $\kappa$*-contractive and hence is a* simple *SCM.*

*Proof.* **(1) Composition reduces to a single affine map.** For any coordinate $j \in \mathcal{I}$ the successive updates act as $x_j \mapsto a_j^{(m)}\big(a_j^{(m-1)}(\ldots a_j^{(1)} f_j(x, e) + b_j^{(1)} \ldots) + b_j^{(m-1)}\big) + b_j^{(m)}$, which can be written $a_j^{\mathrm{comp}} f_j(x, e) + b_j^{\mathrm{comp}}$ with $a_j^{\mathrm{comp}} := \prod_{r: \ j \in \tilde{\mathcal{I}}^{(r)}} a_j^{(r)}$ and a corresponding affine drift $b_j^{\mathrm{comp}}$. Since each factor satisfies $|a_j^{(r)}| \leq 1$, we have $|a_j^{\mathrm{comp}}| \leq 1$. Hence the final system coincides with $g(x, e) = a^{\mathrm{comp}} \odot f(x, e) + b^{\mathrm{comp}}$.

**(2) Contraction constant is preserved.** Let $D := \mathrm{diag}(a^{\mathrm{comp}})$ denote the diagonal matrix of multiplicative factors. For any $u \in \mathbb{R}^{|\mathcal{I}|}$, $\|D u\|_p \leq \|u\|_p$ because every diagonal entry satisfies $|a_j^{\mathrm{comp}}| \leq 1$. Hence for all $x, y$ and any $e$

$$\|g(x, e) - g(y, e)\|_p = \|D\big(f(x, e) - f(y, e)\big)\|_p \leq \|f(x, e) - f(y, e)\|_p \leq \kappa \|x - y\|_p.$$

Thus $g$ inherits the same global contraction constant $\kappa$.

**(3) Simplicity after intervention.** Because the intervened mechanism is still $\kappa$-contractive with $\kappa < 1$, Theorem 1 applies verbatim: the intervened model is uniquely solvable with respect to every subset of endogenous variables, i.e. it is simple. $\qquad\square$

**Remark 1** (Scale factors exceeding one)**.** *The proof of Proposition 1 used the bound* $|a_j| \leq 1$ *to conclude that the diagonal scaling* $D = \mathrm{diag}(a_j)$ *obeys* $\|Du\|_p \leq \|u\|_p$ *and therefore does not enlarge the global Lipschitz constant. If some multipliers satisfy* $|a_j| > 1$*, simplicity can still hold, but one must verify an additional criterion:*

*Let* $\kappa < 1$ *be the original contraction constant of* $f$ *and define*

$$\kappa_{\mathrm{max}} := \big(\max_{j \in \tilde{\mathcal{I}}} |a_j|\big) \kappa.$$

*Because* $\|D\|_p = \max_j |a_j|$*, the intervened mechanism is globally* $\kappa_{\mathrm{max}}$*-Lipschitz:* $\|Df(x, e) - Df(y, e)\|_p \leq \kappa_{\mathrm{max}} \|x - y\|_p$*. If* $\kappa_{\mathrm{max}} < 1$ *the model remains a contraction and is therefore simple; if* $\kappa_{\mathrm{max}} \geq 1$ *the contraction proof no longer ensures uniqueness and additional analysis is required.*

**Proposition 2** (Sub-Gaussian Tails under Gaussian Noise for Shift–Scale Counterfactuals).
*Let the SCM $\mathcal{M} = \langle \mathcal{I}, \mathcal{J}, \mathcal{X}, \mathcal{E}, f, \mathbb{P}_{\mathcal{E}} \rangle$ satisfy the global $\ell^2$-contraction condition of Theorem 1 with constant $\kappa < 1$. Apply any shift–scale intervention $\mathrm{ss}(\tilde{\mathcal{I}}, a_{\tilde{\mathcal{I}}}, b_{\tilde{\mathcal{I}}})$ to the twin SCM, with $\max_{j \in \tilde{\mathcal{I}}} |a_j| \leq 1$, and let $(\mathbf{X}, \mathbf{X}')$ be the unique endogenous solution (Definition 2) of the intervened twin model. Assume*

*(a)* **1-Lipschitz in noise (in $\ell^2$):** *For all $x \in \mathcal{X}$, $e_1, e_2 \in \mathcal{E}$,*
$$\|f(x, e_1) - f(x, e_2)\|_2 \leq \|e_1 - e_2\|_2.$$

*(b)* **Gaussian noise:** *$\mathbf{E} \sim \mathcal{N}(\mu, \Sigma)$ with $\Sigma \preceq \sigma^2 I$ for some $\sigma^2 > 0$.*

*Then for every 1-Lipschitz functional $h : \mathcal{X} \times \mathcal{X} \to \mathbb{R}$ (w.r.t. $\ell^2$),*
$$\mathbb{P}\Big(h(\mathbf{X}, \mathbf{X}') - \mathbb{E}h(\mathbf{X}, \mathbf{X}') \geq t\Big) \leq \exp\Big(-\tfrac{t^2}{4(1-\kappa)^{-2}\sigma^2}\Big), \qquad t > 0.$$

*Hence $(\mathbf{X}, \mathbf{X}')$ is sub-Gaussian with proxy $2(1-\kappa)^{-2}\sigma^2$.*

*Proof.* Let $\Phi : \mathcal{E} \to \mathcal{X} \times \mathcal{X}$ denote the unique measurable solution map of the shift–scale–intervened twin SCM. That is,
$$(\mathbf{X}, \mathbf{X}') = \Phi(\mathbf{E}),$$
where $\mathbf{E} \sim \mathbb{P}_{\mathcal{E}}$ is the exogenous random variable. By the global $\kappa$-contraction in $x$ and the 1-Lipschitz property in $e$ (both in $\ell^2$), $\Phi$ is $L$-Lipschitz with constant $L := \frac{\sqrt{2}}{1-\kappa}$ in $\ell^2$:

$$\|\Phi(e_1) - \Phi(e_2)\|_2 \leq \frac{\sqrt{2}}{1-\kappa}\|e_1 - e_2\|_2 \quad \text{for all } e_1, e_2 \in \mathcal{E}.$$

Take any $h : \mathcal{X} \times \mathcal{X} \to \mathbb{R}$ that is 1-Lipschitz in $\ell^2$. Write $\mathbf{E} = \mu + \Sigma^{1/2}\mathbf{Z}$ with $\mathbf{Z} \sim \mathcal{N}(0, I)$.

Then $g(z) := h\big(\Phi(\mu + \Sigma^{1/2}z)\big)$ is $L\|\Sigma^{1/2}\|_2$-Lipschitz in $\ell^2$. Since $\Sigma \preceq \sigma^2 I$, we have $\|\Sigma^{1/2}\|_2 \leq \sigma$. By Gaussian concentration for Lipschitz functions (e.g., (Vershynin, 2018, Thm. 5.2.3)),

$$\mathbb{P}\Big(h(\mathbf{X}, \mathbf{X}') - \mathbb{E}[h(\mathbf{X}, \mathbf{X}')] \geq t\Big) \leq \exp\Big(-\frac{t^2}{2L^2\sigma^2}\Big) = \exp\Big(-\frac{t^2}{4(1-\kappa)^{-2}\sigma^2}\Big).$$

This proves the stated concentration inequality. $\qquad\square$

**Remark 2** ($\ell^p$ version of Proposition 2). *Let all Lipschitz and contraction conditions in Proposition 2 be measured in $\ell^p$ ($1 \leq p \leq \infty$), and keep the Gaussian noise assumption $\mathbf{E} \sim \mathcal{N}(\mu, \Sigma)$. Then for every 1-Lipschitz $h : (\mathcal{X} \times \mathcal{X}, \|\cdot\|_p) \to \mathbb{R}$ and all $t > 0$,*

$$\mathbb{P}\Big(h(\mathbf{X}, \mathbf{X}') - \mathbb{E}h(\mathbf{X}, \mathbf{X}') \geq t\Big) \leq \exp\Big(-\frac{t^2}{2^{1+\frac{2}{p}}(1-\kappa)^{-2}\|\Sigma^{1/2}\|_{2\to p}^2}\Big),$$

*where $\|A\|_{2\to p} := \sup_{x \neq 0} \frac{\|Ax\|_p}{\|x\|_2}$. In particular, if $\Sigma \preceq \sigma^2 I$ and $d := \dim(\mathcal{E})$, then*

$$\|\Sigma^{1/2}\|_{2\to p} \leq \sigma \|I\|_{2\to p} \quad \text{with} \quad \|I\|_{2\to p} = \begin{cases} 1, & p \geq 2, \\ d^{\frac{1}{p}-\frac{1}{2}}, & 1 \leq p < 2, \end{cases}$$

*so the proxy equals $2^{\frac{2}{p}}(1-\kappa)^{-2}\sigma^2$ for $p \geq 2$, and $2^{\frac{2}{p}}(1-\kappa)^{-2}\sigma^2 d^{2(\frac{1}{p}-\frac{1}{2})}$ for $1 \leq p < 2$. Proof sketch. Write $\mathbf{E} = \mu + \Sigma^{1/2}\mathbf{Z}$, $\mathbf{Z} \sim \mathcal{N}(0, I)$; then $z \mapsto h(\Phi(\mu + \Sigma^{1/2}z))$ is $2^{\frac{1}{p}}(1-\kappa)^{-1}\|\Sigma^{1/2}\|_{2\to p}$-Lipschitz in $\ell^2$, and Gaussian Lipschitz concentration applies.*

## 4  Example

Consider a linear cyclic SCM with mutual dependence between consumption ($C$) and income ($I$):

$$\begin{aligned} C &= 0.50\,I + 1 + E_C, \\ I &= 0.40\,C + 0.50 + E_I, \end{aligned} \qquad (E_C, E_I)^\top \sim \mathcal{N}(\mathbf{0},\, 0.04\,\mathbf{I}_2). \qquad (2)$$

Written in matrix form $X = (C, I)^\top$, $X = AX + b + E$ with

$$A = \begin{pmatrix} 0 & 0.50 \\ 0.40 & 0 \end{pmatrix}, \qquad b = \begin{pmatrix} 1 \\ 0.5 \end{pmatrix}, \qquad E = (E_C, E_I)^\top.$$

Because the spectral norm is $\|A\|_2 \le \|A\|_F = \sqrt{0.25 + 0.16} = 0.6403 < 1$, the model is *globally contractive* and therefore *simple* in the sense of Definition 5.

For any $2 \times 2$ matrix $A = \left[\begin{smallmatrix} 0 & a \\ b & 0 \end{smallmatrix}\right]$ with $ab < 1$, $(I - A)^{-1} = \frac{1}{1-ab}\left[\begin{smallmatrix} 1 & a \\ b & 1 \end{smallmatrix}\right]$. With $(a, b) = (0.50, 0.40)$ we have $(I - A)^{-1} = \left[\begin{smallmatrix} 1.25 & 0.625 \\ 0.50 & 1.25 \end{smallmatrix}\right]$ and therefore

$$\mathbb{E}[X] = (I - A)^{-1}b = (1.5625, \ 1.125)^\top, \tag{3}$$

$$\mathrm{Cov}(X) = (I - A)^{-1}(0.04\,\mathbf{I}_2)(I - A)^{-\top} = \begin{pmatrix} 0.0781 & 0.0563 \\ 0.0563 & 0.0725 \end{pmatrix}. \tag{4}$$

Thus $X_{\mathrm{obs}} \sim \mathcal{N}(\mu, \Sigma)$ with $\mu$ and $\Sigma$ given in (3)–(4). The correlation between $C$ and $I$ is $\rho = 0.75$.

**Shift–Scale intervention on $I$.**  Suppose a fiscal policy reform dampens the effect of both consumption and random shocks on income by a factor $\alpha = 0.8$, and provides a fixed income supplement of $\beta = 1.0$ units:

$$\mathrm{ss}(I, \alpha, \beta) \ : \ I \ \leftarrow \ \alpha(0.40\,C + 0.50 + E_I) + \beta, \quad \alpha = 0.8, \ \beta = 1.0.$$

The intervened structural parameters are

$$A' = \begin{pmatrix} 0 & 0.50 \\ 0.32 & 0 \end{pmatrix}, \qquad b' = \begin{pmatrix} 1 \\ 1.4 \end{pmatrix}, \quad \text{so} \quad \|A'\|_2 \le \|A'\|_F = \sqrt{0.25 + 0.1024} = 0.5936 < 1.$$

Scaling also affects the exogenous term: $E' = (E_C, \alpha E_I)^\top$, $\Sigma_{E'} = \mathrm{diag}(0.04, \alpha^2 \cdot 0.04) = \mathrm{diag}(0.04, 0.0256)$. Contractivity is preserved, hence a unique *interventional* equilibrium exists:

$$\mathbb{E}[X_{\mathrm{int}}] = (I - A')^{-1}b' = (2.024, \ 2.048)^\top, \tag{5}$$

$$\mathrm{Cov}(X_{\mathrm{int}}) = (I - A')^{-1}\Sigma_{E'}(I - A')^{-\top} = \begin{pmatrix} 0.0658 & 0.0363 \\ 0.0363 & 0.0421 \end{pmatrix}, \qquad \rho_{\mathrm{int}} \approx 0.69. \tag{6}$$

Consumption rises by $\sim 29\,\%$ and income by $\sim 82\,\%$, while the $C$–$I$ correlation falls from $0.75$ to $0.69$.

**From intervention to *counterfactual* via the twin SCM.**  Equations (5)–(6) describe the *interventional* distribution $P(X_{\mathrm{int}})$, i.e. what we would observe *if* the shift–scale policy were enacted *for the whole population*. To answer individual–level *counterfactual* queries ("what would *this* household's consumption be had the policy applied?") we follow the *twin network* construction. Let $(c, i)$ denote the *actually observed* consumption and income for one household. The twin-SCM duplicates every endogenous variable and shares the same exogenous noise:

$$\boxed{\begin{aligned} C &= 0.50\,I + 1 + E_C, & C' &= 0.50\,I' + 1 + E_C, \\ I &= 0.40\,C + 0.50 + E_I, & I' &= 0.8(0.40\,C' + 0.50 + E_I) + 1. \end{aligned}} \tag{Twin SCM}$$

where the primed copy encodes the shift–scale intervention $\mathrm{ss}(I, \alpha{=}0.8, \beta{=}1.0)$ and the unprimed copy remains factual. From the factual equations, we can solve

$$E_C = c - 0.50\,i - 1, \qquad E_I = i - 0.40\,c - 0.50. \tag{7}$$

Eliminating $C'$ from the primed equations of the twin SCM yields $0.84\,I' = 1.72 + 0.8\,E_I + 0.32\,E_C$. Substituting 7 gives the counterfactual income $I'(c, i) = \frac{25}{21} + \frac{16}{21}i = 1.190476 + 0.761905\,i$. Back-substitution furnishes the *counterfactual consumption* $C'(c, i) = c + \frac{25}{42} - \frac{5}{42}i = c + 0.595238 - 0.119048\,i$. Above equations give the *counterfactual response mapping* $(c, i) \mapsto (C', I')$. Because the mapping is affine and the factual distribution is Gaussian, the *marginal counterfactual* $(C', I')$ is again Gaussian with $\mathbb{E}[(C', I')^\top] = (2.024, 2.048)^\top$ and covariance exactly matching (6). Here, the twin-SCM reconciles individual counterfactual semantics $(C'(c, i), I'(c, i))$ with the *population-level* interventional distribution already reported, while remaining in closed form due to the linearity and contractiveness of the model.

## 5    Limitations

The $\kappa$–contraction condition must hold *uniformly* over the entire state space; many realistic feedback systems may violate this in certain regimes, even though they still admit unique equilibria. Our concentration result further relies on Gaussianity of the exogenous variables. For heavy-tailed or merely bounded-moment noise, one typically obtains polynomial rather than exponential concentration, which we do not analyze here. We also invoke Banach's theorem on closed (and hence complete) coordinate domains; models whose natural domains are open or lie on manifolds require additional care. From a practical standpoint, certifying global Lipschitz constants of black-box simulators is challenging; data-driven or local contraction diagnostics may be more feasible in applications. Finally, our closure results explicitly cover only shift–scale maps with bounded gains ($|a_j| \leq 1$); interventions with larger multiplicative factors, stochastic policies, or more general functional forms are not yet addressed.

At present, our analysis is restricted to *shift–scale interventions*. This class, however, already strictly generalizes hard (do-) interventions and provides a principled foundation for broader extensions. The key theoretical principle underpinning our guarantees is the preservation of *global contractivity*—that is, the global Lipschitz constant of the intervened system remains below one. This property ensures unique solvability of the intervened SCM, even in the presence of cycles, and is abstract enough to accommodate richer intervention types, provided they do not destroy contractivity. Consequently, our results extend directly to any intervention family (including nonlinear, stochastic, or more complex parametric changes) that preserves the contraction property. An explicit analysis of broader classes of interventions, and sufficient conditions under which they preserve contractivity, is a promising direction for future work

## 6    Conclusion and Future Work

We have established a principled foundation for counterfactual inference in cyclic structural causal models under shift–scale interventions. Leveraging a global contraction assumption, we proved that such models are simple, ensuring unique solvability even in the presence of feedback. We further showed that shift–scale interventions preserve solvability, are closed under composition, and admit sub-Gaussian tail bounds for counterfactual functionals under natural regularity assumptions. These results demonstrate that contraction-based SCMs offer a mathematically tractable yet expressive class for reasoning about interventions and counterfactuals in cyclic settings. Future research may focus on developing deep generative models for cyclic counterfactuals, leveraging the theoretical foundations established here.

## Acknowledgments and Disclosure of Funding

This research was funded in part by the Indo-French Centre for the Promotion of Advanced Research (IFCPAR/CEFIPRA) through project number CSRP 6702-2.

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

# A  Supplementary Material

## A.1  Equivalence of Twin SCM and Pearl's Abduction-Action-Prediction (AAP) Procedure

> **Theorem 3.** *Let $\mathcal{M} = \langle \mathcal{I}, \mathcal{J}, \mathcal{X}, \mathcal{E}, f, \mathbb{P}_{\mathcal{E}} \rangle$ be a structural causal model that is* simple *(i.e., uniquely solvable with respect to every subset of endogenous variables). Assume:*
>
> - *(i) the exogenous vector $\mathbf{E} \sim \mathbb{P}_{\mathcal{E}}$ has a joint density and mutually independent components;*
> - *(ii) the counterfactual query is defined by a perfect intervention $\mathrm{do}(X_{\mathcal{A}} := \tilde{x}_{\mathcal{A}})$ for some $\mathcal{A} \subseteq \mathcal{I}$.*
>
> *Let $\mathbf{X} \sim \mathcal{M}$ denote the factual world, and define the counterfactual outcome via:*
>
> - ***Twin SCM definition:*** *Solve the intervened twin SCM where the do-intervention is applied to the primed copy, yielding random variables $(\mathbf{X}, \mathbf{X}')$.*
> - ***Pearl's AAP (Pearl, 2009) procedure:*** *Sample $\mathbf{E} \sim \mathbb{P}_{\mathcal{E}}$ conditioned on $\mathbf{X} = x^{\mathrm{obs}}$, and solve the intervened SCM with this fixed noise to obtain $\mathbf{X}^{cf}$.*
>
> *Then, the counterfactual distribution derived from the **twin SCM** coincides with the distribution obtained from **Pearl's abduction–action–prediction (AAP)** procedure:*
>
> $$\mathbb{P}_{\mathbf{X}'|\mathbf{X}=x^{\mathrm{obs}}} = \mathbb{P}_{\mathbf{X}^{\mathrm{cf}}|\mathbf{X}=x^{\mathrm{obs}}}.$$

*Proof.* Since $\mathcal{M}$ is simple, by definition there exists a unique measurable function $g : \mathcal{E} \to \mathcal{X}$ such that for every $e \in \mathcal{E}$,

$$g(e) = f(g(e), e), \quad \text{so that } \mathbf{X} = g(\mathbf{E}) \text{ a.s.}$$

Similarly, after the intervention $\mathrm{do}(X_{\mathcal{A}} := \tilde{x}_{\mathcal{A}})$, the intervened SCM admits a unique measurable solution map $g^{\mathrm{do}} : \mathcal{E} \to \mathcal{X}$ (Proposition 3.8, Bongers et al. (2021)).

In the twin SCM construction, both copies (factual and counterfactual) share the same exogenous vector $\mathbf{E}$, and the intervention is applied only to the first copy. Solving the twin SCM yields:

$$(\mathbf{X}, \mathbf{X}') = (g(\mathbf{E}), \ g^{\mathrm{do}}(\mathbf{E})).$$

Let $x^{\mathrm{obs}} \in \mathcal{X}$ be the observed factual outcome. Pearl's procedure involves:

- *Abduction:* condition on the observation $\mathbf{X} = x^{\mathrm{obs}}$, which corresponds to conditioning on the set $\{e \in \mathcal{E} : g(e) = x^{\mathrm{obs}}\}$;

- *Action:* apply the intervention to obtain the new function $g^{\mathrm{do}}$;

- *Prediction:* evaluate $\mathbf{X}^{\mathrm{cf}} = g^{\mathrm{do}}(\mathbf{E})$ using the same noise, but now drawn from the posterior $\mathbb{P}_{\mathcal{E}|g(\mathbf{E})=x^{\mathrm{obs}}}$.

Define the posterior distribution on exogenous noise:

$$\mu_{x^{\mathrm{obs}}}(A) := \mathbb{P}\big(\mathbf{E} \in A \,\big|\, g(\mathbf{E}) = x^{\mathrm{obs}}\big), \qquad \text{for all measurable } A \subseteq \mathcal{E}.$$

This measure is supported on the set

$$\{e \in \mathcal{E} : g(e) = x^{\mathrm{obs}}\}.$$

**Equality of counterfactual laws.** By construction, $\mathbf{X}' = g^{\mathrm{do}}(\mathbf{E})$, so the counterfactual distribution from the twin SCM (given $\mathbf{X} = x^{\mathrm{obs}}$) is:

$$\mathbb{P}_{\mathbf{X}'|\mathbf{X}=x^{\mathrm{obs}}}(B) = \mu_{x^{\mathrm{obs}}}\big(e : g^{\mathrm{do}}(e) \in B\big), \qquad B \subseteq \mathcal{X}.$$

Pearl's method also conditions on $\{\mathbf{X} = x^{\mathrm{obs}}\}$, inducing the same posterior $\mu_{x^{\mathrm{obs}}}$ over exogenous variables. Then, the counterfactual outcome is $\mathbf{X}^{\mathrm{cf}} = g^{\mathrm{do}}(\mathbf{E})$ with $\mathbf{E} \sim \mu_{x^{\mathrm{obs}}}$. Thus,

$$\mathbb{P}_{\mathbf{X}^{\mathrm{cf}}|\mathbf{X}=x^{\mathrm{obs}}}(B) = \mu_{x^{\mathrm{obs}}}\big(e : g^{\mathrm{do}}(e) \in B\big),$$

which is the same expression as for the twin SCM. $\qquad\square$

The counterfactual law derived from the twin SCM coincides with the one obtained from Pearl's abduction–action–prediction procedure under unique solvability.

**Remark 3** (Notation $\mathbf{X} \sim \mathcal{M}$). *The shorthand $\mathbf{X} \sim \mathcal{M}$ means that the random vector $\mathbf{X}$ is a solution of the SCM $\mathcal{M}$.*

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
