# OpenReview forum: "Cyclic Counterfactuals under Shift–Scale Interventions"
_NeurIPS.cc/2025/Conference — NeurIPS 2025 poster_

### Official Review · Reviewer_Ucnx · 2025-06-20

**Clarity:** 4
**Significance:** 2
**Originality:** 3
**Rating:** 5
**Confidence:** 4

**Summary:**

The paper focuses on cyclic SCMs and more precisely on their behavior under a type of soft interventions, the shift-scale interventions do(aX+b). A number of theoretical results is proved, including the unique solvability of twin SCMs, used for modelling counterfactual distributions, under ss interventions. The latter implies that counterfactual distributions associated with such interventions are well-defined for cyclic SCMs.

**Questions:**

Q1. What are the key differences between Theorem 2 from your paper and Proposition 8.2 from Bogers et.al.?

Q2. Could I kindly ask the authors to explain what P-almost surely means in Definition 2? It could be helpful if you add an explanation in the paper for completeness.

Q3. Could you elaborate on the last sentence from the paragraph in Section 5?

**Ethical Concerns:**

["NO or VERY MINOR ethics concerns only"]

**Final Justification:**

The authors promised to include the comparison I asked.

**Limitations:**

Limitations are mentioned. Maybe some more discussion on the challenges that other types of soft interventions pose would be helpful.

**Paper Formatting Concerns:**

All good

**Quality:**

3

**Strengths And Weaknesses:**

+ The paper is clearly written, and results are well explained.
+ I read the proofs of the two theorems only. Their exposition was nice, and I found them easy to follow. The proofs seemed correct.
+ The field of cyclic SCMs is indeed under explored and important. Thus, efforts to shed more light in this domain have value.

- I found the connection of this paper to prior work poor. I strongly advise the authors to: (1) mention previous results from Bongers et al in cyclic SCMs, and especially their results related to hard interventions; (2) highlight any potential overlap in methodology and draw a connection to this paper and your explicitly mentioning the extra challenges you had to face in order to prove your results. See my questions for some examples

- The contributions of the paper are rather limited. It would be interested to see a similar study for other subfamilies of soft interventions and/or the case with more than one simultaneous soft interventions. The latter produces counterfactual distributions which might include more than two counterfactual worlds and hence cannot be modeled by twin SCMs.

---

> ### Author Rebuttal · Authors · 2025-07-30
>
> We sincerely thank the reviewer for their careful reading and thoughtful, constructive feedback.
>
>
>
>
> The contributions of the paper are rather .... two counterfactual worlds and hence cannot be modeled by twin SCMs.
> > Our current work is restricted to shift–scale interventions, but this class already strictly generalizes hard interventions and serves as a foundation for further extensions. The core theoretical principle underpinning our results is the preservation of global contractivity (i.e., the global Lipschitz constant remains below one) after intervention. This property ensures unique solvability of the intervened SCM, even in the presence of cycles, and is abstract enough to accommodate more general intervention types, provided they do not destroy contractivity. Thus, our results can be directly extended to any class of interventions (including nonlinear, stochastic, or more complex parametric changes) that preserve this contraction property. An explicit analysis of broader classes of interventions, and sufficient conditions under which they preserve contractivity, is a promising direction for future work
>
> >  Regarding multiple simultaneous soft interventions, as you have rightly pointed out that classical twin SCMs are limited to two worlds. Extending our fixed-point and contractivity analysis to multi-world counterfactuals is a natural and promising avenue for future research.
>
> ---
>
> I found the connection of this paper to prior work poor. I strongly ... See my questions for some examples.
>
> > Thank you for this valuable suggestion. We will include a discussion of prior results by Bongers et al., particularly their findings on hard interventions. This will help clarify how our work extends these foundational contributions.
>
> > Our responses to your questions provide further clarification on these aspects.
> ---
>
> (Q1)
> > Theorem 2 provides sufficient conditions for unique solvability of the twin SCM under shift–scale interventions. Proposition 8.2 is a closure property: it states that if an SCM is simple, then its twin SCM, marginalizations, and perfect interventions will also be simple. It does not provide a sufficient condition to ensure simplicity (e.g., contraction); rather, it relies on the SCM being already simple.
>
> ---
>
> (Q2)
> > It means that the structural equations hold with probability 1 under the probability measure $\mathbb{P}$ defined on the underlying probability space. In other words, the set of  elements of the sample space $\omega \in \Omega$ for which the equation does not hold has probability zero.
> "Almost surely" (abbreviated as a.s.) is a standard concept in probability theory.
>
>
> ---
>
> (Q3)
> > We have rephrased the sentence as: "Future research may focus on developing deep generative models for cyclic counterfactuals, leveraging the theoretical foundations established here".
> Note that the majority of counterfactual inference approaches using deep generative models restrict the causal structure to directed acyclic graphs (DAGs).
>
>
> We hope our responses effectively address the reviewer’s concerns and respectfully ask that you consider raising  your score based on these clarifications.

---

> > ### Comment · Reviewer_Ucnx · 2025-08-04
> >
> > Thank you for your answer. I am raising my score to accept. Please do add the comparison with prior work as promised.

---

> > > ### Author Response · Authors · 2025-08-05
> > >
> > > Thank you very much. We will incorporate that discussion.

---

### Official Review · Reviewer_vCep · 2025-07-01

**Clarity:** 1
**Significance:** 3
**Originality:** 3
**Rating:** 5
**Confidence:** 2

**Summary:**

This paper addresses the challenge of counterfactual inference in cyclic structural causal models (SCMs), which better reflect the feedback loops and cyclical dependencies found in many real-world systems such as biological networks. While traditional counterfactual methods assume acyclic models (DAGs), this work departs from that norm and explores shift-scale interventions within cyclic SCMs. The authors propose a new framework for conducting counterfactual analysis under these conditions and demonstrate its theoretical soundness and practical utility, laying groundwork for more realistic causal reasoning in complex systems.

**Questions:**

1. Which part of the paper explains how the proposed method handles cyclic structures?
2. Could the authors elaborate where the twin-SCM construction interacts with the cyclic structure?
3. What is the computational cost of solving the (shift-scale) twin SCM in high dimensions, and are there efficient algorithms?
4. Can the authors provide simulations or real-data case studies to demonstrate (a) cycles in action, (b) contraction diagnostics, and (c) counterfactual estimation under shift-scale interventions?

**Ethical Concerns:**

["NO or VERY MINOR ethics concerns only"]

**Final Justification:**

The authors have adequately addressed my concerns. I will raise my score to accept.

**Limitations:**

Yes

**Quality:**

2

**Strengths And Weaknesses:**

Strengths
- Addresses a major gap by extending counterfactual inference to cyclic SCMs, which better model feedback loops in real systems (e.g., biology).

Weaknesses
- The paper never clearly highlights which sections or results are specifically responsible for handling cyclic structures, making it harder for readers to identify where cycles are addressed.
- The paper lacks empirical validation. No simulations or real-data experiments are provided to illustrate practical performance.
- Computational aspects (e.g., how to verify contraction or solve high-dimensional twin SCMs) are not discussed, which could hinder adoption in large-scale systems.

---

> ### Author Rebuttal · Authors · 2025-07-30
>
> We are grateful to the reviewer for careful reading and constructive comments offered.
>
> (W1) & (Q1,Q2)
> >  We recognize that the literature on counterfactual inference has predominantly focused on acyclic structural causal models (SCMs). It is therefore essential to clarify how our work generalizes these results to the cyclic setting. Our paper does not assume acyclicity. Rather, a major motivation and contribution of our work is to establish a rigorous theoretical framework for counterfactual inference in cyclic SCMs. All our theoretical results, proofs, and constructions are formulated for general SCMs
> that may be cyclic. The acyclic (DAG) case is simply a special instance.
>
> ---
> (W2) & (Q2, Q4)
> > We appreciate the reviewer’s concern. We have added a detailed worked example to the revised manuscript
>
> ## **Example:**
>  > Consider a linear cyclic SCM with mutual dependence between consumption ( $C$ ) and income ( $I$ ):
> \begin{equation}
> \begin{aligned}
> C = 0.50\,I + 1 + E_{C}, \quad
> I = 0.40\,C + 0.50 + E_{I},
> \end{aligned}
> \qquad
> (E_{C},E_{I})^\top \sim \mathcal N(\mathbf 0,\,0.04\,\mathbf I_2).
> \end{equation}
> Written in matrix form  $X =(C,I)^\top$, $X = A X + b + E$ with
> \begin{align*}
> A = \begin{pmatrix} 0 & 0.50 \\\ 0.40 & 0 \end{pmatrix}, \qquad
> b = \begin{pmatrix} 1 \\\ 0.5 \end{pmatrix}, \qquad
> E = (E_C, E_I)^\top.
> \end{align*}
> Because the spectral norm is $\|A\|_2 = \sqrt{0.50\times0.40}=0.447 < 1$,  the model is globally contractive and therefore simple in the sense of Definition 4.
> For any $2{\times}2$ matrix $A=\bigl[\begin{smallmatrix}0 & a\\\ b & 0\end{smallmatrix}\bigr] $ with \(ab<1\),
> $
> (I-A)^{-1}= \frac{1}{1-ab}\bigl[\begin{smallmatrix}1 & a\\\ b & 1\end{smallmatrix}\bigr].
> $ With \\((a,b)=(0.50,0.40)\\) we have
> \\((I-A)^{-1}=%
> \bigl[\begin{smallmatrix}1.25 & 0.625\\\ 0.50 & 1.25\end{smallmatrix}\bigr]\\) and therefore
> \begin{align*}
> \mu = \mathbb E[X] = (I-A)^{-1} b
>             = (1.5625,\;1.125)^\top, \\\
> \Sigma = \operatorname{Cov}(X) = (I-A)^{-1}(0.04\,\mathbf I_2)(I-A)^{-\top}
>             =\begin{pmatrix}0.0781 & 0.0563\\\ 0.0563 & 0.0725\end{pmatrix}.
> \end{align*}
>
> > Thus  $ X_{\text{obs}}\sim\mathcal N\ \bigl(\mu,\Sigma\bigr) $ with $ \mu $ and $ \Sigma $ .  The correlation between \\(C\\) and \\(I\\) is \\(\rho=0.75\\).
>
> > **Shift–scale intervention.**
> Suppose a fiscal policy reform dampens the effect of both consumption and random shocks on income by a factor $\alpha=0.8$, and provides a fixed income supplement of $\beta=1.0$ units:
> \\[
> \mathrm{ss}(I,\alpha,\beta) :
> I \leftarrow \alpha\bigl(0.40\,C + 0.50 + E_{I}\bigr) + \beta,
> \quad
> \alpha=0.8, \beta=1.0.
> \\]
> The intervened structural parameters  are
> \begin{align*}A'=\begin{pmatrix}0 & 0.50 \\\ 0.32 & 0\end{pmatrix}, \qquad
> b'=\begin{pmatrix}1 \\\ 1.4\end{pmatrix},
> \quad \text{so}\quad
> \\|A'\\|_2=\sqrt{0.50\times0.32}=0.400<1.
> \end{align*}
>
>
> > Scaling also affects the exogenous term:
> \\(
> E'=(E_{C},\,\alpha E_{I})^\top,
> \Sigma_{E'}=\operatorname{diag}(0.04,\alpha^{2} \\cdot 0.04)
>            =\operatorname{diag}(0.04,0.0256).
> \\)
>
>
>
> > Contractivity is preserved, hence a unique interventional equilibrium exists:
> \begin{align*}
> \mathbb E[X_{\text{int}}] =(I-A')^{-1} b'
>         =(2.024,\;2.048)^\top,\\\
> \operatorname{Cov}(X_{\text{int}}) =(I-A')^{-1}\Sigma_{E'}(I-A')^{-\top}
>   =\begin{pmatrix} 0.0658 & 0.0363 \\\ 0.0363 & 0.0421\end{pmatrix},
> \qquad
> \rho_{\text{int}} \approx 0.69.
> \end{align*}
> Consumption rises by \\(\sim\ 29\\%\\) and income by \\(\sim 82 \\%\\), while the
> \\(C\\)–\\(I\\) correlation falls from \\(0.75\\) to \\(0.69\\).
>
>
> > **From intervention to counterfactual via the twin SCM.**
> Above equations describe the interventional distribution $P(X_{\text{int}})$, i.e. what we would observe if the shift–scale policy were enacted for the whole population. To answer individual–level counterfactual queries (``what would this household’s consumption be had the policy applied?’’) , we follow the twin network construction:
> Let $(c,i)$ denote the actually observed consumption and income for one household.  The twin‑SCM duplicates every endogenous variable and shares the same exogenous noise:
>
> \\[
> \boxed{
> \begin{aligned}
> C &= 0.50 I + 1 + E_{C},              &  C' &= 0.50I' + 1 + E_{C},\\\
> I &= 0.40C + 0.50 + E_{I},           &  I' &= 0.8\bigl(0.40C' + 0.50 + E_{I}\bigr) + 1.
> \end{aligned}}
> \\]
>
> > where the primed copy encodes the shift–scale intervention
> $\mathrm{ss}(I,\alpha{=}0.8,\beta{=}1.0)$ and the unprimed copy remains
> factual. From the factual equations, we can solve
> \begin{equation}
> E_{C}=c-0.50i-1,
> \qquad
> E_{I}=i-0.40c-0.50.
> \end{equation}
>
> > Eliminating $C'$ from the primed equations of the twin SCM yields
> \\(
> 0.84I' = 1.72 + 0.8E_{I} + 0.32E_{C}.
> \\)
> Substituting $E_{C}$ and $E_{I}$ gives the counterfactual income \\(
> I'(c,i) = \frac{25}{21} + \frac{16}{21}i
>         = 1.190476 + 0.761905i.
> \\) Back‑substitution furnishes the counterfactual consumption
> \\(
> C'(c,i) = c + \tfrac{25}{42} - \tfrac{5}{42}i
>         = c + 0.595238 - 0.119048i. \\)
>
>
> > Above equations give the counterfactual response mapping $(c,i)\mapsto(C',I')$. Because the mapping is affine and the factual distribution $X_{\mathrm{obs}}\sim\mathcal N(\mu,\Sigma)$
> is Gaussian, the marginal counterfactual $(C',I')$ is again Gaussian with
> \\(\mathbb E[(C',I')^\top]=(2.024,2.048)^\top\\) and covariance $\begin{pmatrix} 0.0658 & 0.0363 \\\ 0.0363 & 0.0421\end{pmatrix}$.
> Here, the twin‑SCM reconciles individual counterfactual semantics
> \\((C'(c,i),I'(c,i))\\) with the population‑level interventional
> distribution already reported, while remaining in closed form due to the linearity and contractiveness of the model.
>
>
>
> ---
> (W3) & (Q3)
> > Our work is intended as a foundational contribution establishing well-posedness and unique solvability for counterfactual queries in cyclic SCMs under shift–scale interventions, rather than as a recipe for scalable counterfactual estimation. We do not provide a general-purpose algorithm for estimating counterfactuals or for solving high-dimensional twin SCMs.
> The solution to the twin SCM (and thus the counterfactuals) can always, in principle, be obtained by iterating the contraction map (Banach fixed-point iteration), which is well known to converge geometrically fast for contraction mappings.
>
> >In practice, one may verify contraction via power‑iteration, semidefinite‑program and guarantee it with spectral‑norm or Jacobian penalties already standard in deep learning pipelines. For further details, please refer to:
> *"Efficient and Accurate Estimation of Lipschitz Constants for Deep Neural Networks"*, Fazlyab et al., 2019;
> *"Spectral Normalization for Generative Adversarial Networks"*, Miyato et al., 2018.
> However,  scalability to extremely large systems is outside our current scope and an important direction for future work.
>
>
>
>
> We hope that the additional example addresses the reviewer’s concerns. We kindly request that you reconsider your score in light of these clarifications.

---

> > ### Author Response · Authors · 2025-08-01
> > **Small Correction in the Example**
> >
> > > \begin{aligned} C = 0.50,I + 1 + E_{C}, \quad I = 0.40,C + 0.50 + E_{I}, \\qquad
> > (E_{C},E_{I})^\top \sim \mathcal N(\mathbf 0, ,0.04\mathbf I_2)\end{aligned}
> >
> > should be
> >
> > > \begin{aligned} C = 0.50I + 1 + E_{C}, \quad I = 0.40C + 0.50 + E_{I},  \\qquad
> > (E_{C},E_{I})^\top \sim \mathcal N(\mathbf 0, 0.04\mathbf I_2) \end{aligned}
> >
> > **and**
> >
> > > \\[
> > \mathrm{ss}(I,\alpha,\beta):
> > I \leftarrow \alpha \bigl(0.40\,C + 0.50 + E_{I}\bigr) + \beta,
> > \quad
> > \alpha=0.8,\beta=1.0.
> > \\]
> >
> > should be
> >
> > > \\[
> > \mathrm{ss}(I,\alpha,\beta) :
> > I  \leftarrow  \alpha\bigl(0.40C + 0.50 + E_{I}\bigr) + \beta,
> > \quad
> > \alpha=0.8, \beta=1.0.
> > \\]
> >
> > We apologize for the inconvenience — some extra semicolons and commas are inadvertently included in various places.

---

### Official Review · Reviewer_Na7U · 2025-07-02

**Clarity:** 3
**Significance:** 3
**Originality:** 2
**Rating:** 5
**Confidence:** 2

**Summary:**

Structural causal models are the classical framework in counterfactual inference. However, these models typically require an directed acyclic graph, which excludes the possibility of feedback loops. The article introduces a notion of counterfactuals that generalizes beyond acyclic graphs and can handle cycles. The underlying mathematics is based on contraction mappings and Banach fixed point theory.

**Questions:**

see above.

**Ethical Concerns:**

["NO or VERY MINOR ethics concerns only"]

**Final Justification:**

I keep my score in view of rebuttal and discussion.

**Limitations:**

yes.

**Paper Formatting Concerns:**

No.

**Quality:**

3

**Strengths And Weaknesses:**

Strengths:
- The manuscript is well-written and clear. I enjoyed reading it.
- The manuscript could provide a foundation for follow-up works, as well-posedness of structural causal models with cycles and well-posedness of a corresponding set of interventions is thorougly discussed.

Weaknesses:
- While the mathematical exposition is on point, the main results e.g. Thm.1 and Thm.2 follow in a relatively straightforward manner from Banach fixed point theory. The technical contributions are therefore somewhat limited.
- The authors could expand the discussion on the meaning and interpretations (including a philosophical level) of structural causal models with cyclic graphs. In many physical process there are no feedback effects that are immediate as modelled by the authors. I am thinking of thermodynamic systems, mechanical systems, fluid dynamical systems, etc. that have some sort of "inertia", which means that there is a clear temporal sequence that characterizes how the variables affect each other over time. In such a case modelling with a directed acyclic graph is maybe more appropriate... Can the authors provide simple examples where cyclic graphs are known to occur?

On a similar note, the field of control theory (i.e. the science of feedback) has dealt with feedback loops for a long time although the focus is maybe more on physical and engineering systems, wheras in causal inference the focus lies more on social and economical systems. In control theory there is the notion of well-posedness that has been studied extensively, which is very related to the content of the manuscript (although approached from a different angle). Here is for example a survey on well-posedness for a control-theoretic lens: "Well-posed systems-The LTI case and beyond" by Marius Tucsnak, George Weiss, Automatica, 2014.

---

> ### Author Rebuttal · Authors · 2025-07-30
>
> We thank the reviewer for careful reading and thoughtful comments. We are glad to hear that you enjoyed reading the manuscript.
>
> ---
> (W1)
> >  We agree that the Banach Fixed Point (BFP) theorem underlies our proofs; however, our contribution is not merely a restatement of BFP for cyclic SCMs. The novelty lies in how we specialise it to cyclic structural causal models (SCMs) and in the additional consequences.
> ---
> (W2)
> > -  The term “cyclic” in structural causal models (SCMs) could also be understood as referring to simultaneous structural equations that capture the steady-state behavior of an underlying dynamical system, rather than its transient, time-dependent evolution. One way to interpret these cyclic structural equations is by assuming an underlying discrete-time dynamical system, where the equations act as update rules: the value of each variable at time $t+1$ is computed from the values at time $t$ . The system is then analyzed in the limit as $t\rightarrow \infty$ , focusing on the fixed points to which the dynamics converge.
> Mooij et al. (2013) demonstrate that an alternative, yet natural, interpretation of SCMs emerges when considering systems of ordinary differential equations (ODEs). By examining the equilibrium (steady-state) solutions of such ODEs, one arrives at a structural causal model that is time-independent, but still retains meaningful causal semantics with respect to interventions. Specifically, the semantics of interventions and counterfactuals remain valid and well-defined in this steady-state context, as rigorously formalized by Mooij et al. (2013) and further extended by Bongers et al. (2021). This is by no means the sole pathway to structural causal models; indeed, SCMs may arise through a variety of alternative constructions and representations, depending on the nature of the system under consideration.
>
> *"From Ordinary Differential Equations to Structural Causal Models: the deterministic case", Joris M. Mooij, Dominik Janzing, Bernhard Scholkopf, 2013.*
>
> > **Although many physical processes exhibit inertia, static cyclic structural causal models (SCMs) remain appropriate when we focus on equilibrium behaviour or sample at a temporal resolution coarser than the fastest feedback loop.
>  Examples include gene‑regulatory networks in single‑cell genomics (Rohbeck et al., 2023); market‑equilibrium models (Bongers et al., 2021); predator–prey ecological systems;  Thyroid or reproductive hormone axes exhibit feedback loops (Clarke et al., 2012), etc.**
>
> *"Bicycle: Intervention-Based Causal Discovery with Cycles", Rohbeck et al., 2024.*
>
> *"Modelling mechanisms with causal cycles", Brendan Clarke, Bert Leuridan & Jon Williamson, 2013.*
>
> > - With our limited background in control theory, we note, as you have rightly pointed out, informally that control theorists refer to a feedback interconnection as “well posed” when there exists a unique, stable mapping from inputs to outputs (Tucsnak & Weiss, 2014).  In the context of SCMs, the unique solvability of the structural equations serves an analogous role, ensuring that interventions (do) yield well-defined outcomes.

---

> > ### Comment · Reviewer_Na7U · 2025-08-03
> >
> > I thank the authors for the clarification and keep my score.

---

> > > ### Author Response · Authors · 2025-08-05
> > >
> > > Many thanks.

---

### Official Review · Reviewer_vToH · 2025-07-03

**Clarity:** 3
**Significance:** 3
**Originality:** 3
**Rating:** 5
**Confidence:** 3

**Summary:**

This paper proposes a theoretical framework for counterfactual inference in cyclic structural causal models (SCMs) using shift–scale interventions. While most existing counterfactual frameworks assume acyclic structures (DAGs), many real-world systems, specifically dynamical systems, involve feedback loops. By leveraging a global ℓp-contraction condition, the authors ensure unique solvability of cyclic SCMs even after shift–scale interventions. The paper further establishes closure properties under composition and derives sub-Gaussian concentration bounds for counterfactual outcomes under additional Lipschitz regularity assumptions in the exogenous noise.

**Questions:**

- Could you include a simple, illustrative example of a cyclic SCM to help demonstrate your theoretical results more concretely and make them easier to understand?

- How might your results be used as a foundation to extend the framework to more general types of interventions beyond shift–scale?

**Ethical Concerns:**

["NO or VERY MINOR ethics concerns only"]

**Final Justification:**

The authors’ response addressed my concerns, and the provided example improves the paper’s readability. I will maintain my positive score.

**Limitations:**

It is addressed in the weaknesses.

**Paper Formatting Concerns:**

No concern.

**Quality:**

3

**Strengths And Weaknesses:**

## Strengths
- The paper makes a clear contribution by extending counterfactual reasoning to cyclic causal models.
- The mathematical arguments are carefully detailed and all definitions, theorems, and proofs are clearly presented.

## Weaknesses
- The framework relies heavily on a global contraction condition (k < 1), which is difficult to check or guarantee in practical settings.
- There are no toy examples to help readers understand how the theory plays out in practice.
- The results are limited to shift–scale interventions and a discussion on how the results for these interventions can be generalized to other interventions is missing.

## Minor Comments
- Page 3: The equation between line 83 and 84, 2nd line. ) is not needed.

---

> ### Author Rebuttal · Authors · 2025-07-30
>
> We are grateful to the reviewer for detailed review and helpful comments.
>
>
> (W1)
> > The global contraction is a sufficient condition that lets us invoke Banach’s fixed-point theorem to guarantee unique solvability. Future works may try to relax it to weaker forms.
> In practice, one may verify it via power‑iteration, semidefinite‑program and guarantee it with spectral‑norm or Jacobian penalties already standard in deep learning pipelines.
>
> *"Efficient and Accurate Estimation of Lipschitz Constants for Deep Neural Networks"*, Fazlyab et al., 2019
>
> *"Spectral Normalization for Generative Adversarial Networks"*, Miyato et al., 2018
>
>
> ---
> (W2) & (Q1)
>
> > We have added a detailed worked example to the revised manuscript.
>
> ## **Example:**
>  > Consider a linear cyclic SCM with mutual dependence between consumption ( $C$ ) and income ( $I$ ):
> \begin{equation}
> \begin{aligned}
> C = 0.50\,I + 1 + E_{C}, \quad
> I = 0.40\,C + 0.50 + E_{I},
> \end{aligned}
> \qquad
> (E_{C},E_{I})^\top \sim \mathcal N(\mathbf 0,\,0.04\,\mathbf I_2).
> \end{equation}
> Written in matrix form  $X =(C,I)^\top$, $X = A X + b + E$ with
> \begin{align*}
> A = \begin{pmatrix} 0 & 0.50 \\\ 0.40 & 0 \end{pmatrix}, \qquad
> b = \begin{pmatrix} 1 \\\ 0.5 \end{pmatrix}, \qquad
> E = (E_C, E_I)^\top.
> \end{align*}
> Because the spectral norm is $\|A\|_2 = \sqrt{0.50\times0.40}=0.447 < 1$,  the model is globally contractive and therefore simple in the sense of Definition 4.
> For any $2{\times}2$ matrix $A=\bigl[\begin{smallmatrix}0 & a\\\ b & 0\end{smallmatrix}\bigr] $ with \(ab<1\),
> $
> (I-A)^{-1}= \frac{1}{1-ab}\bigl[\begin{smallmatrix}1 & a\\\ b & 1\end{smallmatrix}\bigr].
> $ With \\((a,b)=(0.50,0.40)\\) we have
> \\((I-A)^{-1}=%
> \bigl[\begin{smallmatrix}1.25 & 0.625\\\ 0.50 & 1.25\end{smallmatrix}\bigr]\\) and therefore
> \begin{align*}
> \mu = \mathbb E[X] = (I-A)^{-1} b
>             = (1.5625,\;1.125)^\top, \\\
> \Sigma = \operatorname{Cov}(X) = (I-A)^{-1}(0.04\,\mathbf I_2)(I-A)^{-\top}
>             =\begin{pmatrix}0.0781 & 0.0563\\\ 0.0563 & 0.0725\end{pmatrix}.
> \end{align*}
>
> > Thus  $ X_{\text{obs}}\sim\mathcal N\ \bigl(\mu,\Sigma\bigr) $ with $ \mu $ and $ \Sigma $ .  The correlation between \\(C\\) and \\(I\\) is \\(\rho=0.75\\).
>
> > **Shift–scale intervention.**
> Suppose a fiscal policy reform dampens the effect of both consumption and random shocks on income by a factor $\alpha=0.8$, and provides a fixed income supplement of $\beta=1.0$ units:
> \\[
> \mathrm{ss}(I,\alpha,\beta) :
> I \leftarrow \alpha\bigl(0.40\,C + 0.50 + E_{I}\bigr) + \beta,
> \quad
> \alpha=0.8, \beta=1.0.
> \\]
> The intervened structural parameters  are
> \begin{align*}A'=\begin{pmatrix}0 & 0.50 \\\ 0.32 & 0\end{pmatrix}, \qquad
> b'=\begin{pmatrix}1 \\\ 1.4\end{pmatrix},
> \quad \text{so}\quad
> \\|A'\\|_2=\sqrt{0.50\times0.32}=0.400<1.
> \end{align*}
>
>
> > Scaling also affects the exogenous term:
> \\(
> E'=(E_{C},\,\alpha E_{I})^\top,
> \Sigma_{E'}=\operatorname{diag}(0.04,\alpha^{2} \\cdot 0.04)
>            =\operatorname{diag}(0.04,0.0256).
> \\)
>
>
>
> > Contractivity is preserved, hence a unique interventional equilibrium exists:
> \begin{align*}
> \mathbb E[X_{\text{int}}] =(I-A')^{-1} b'
>         =(2.024,\;2.048)^\top,\\\
> \operatorname{Cov}(X_{\text{int}}) =(I-A')^{-1}\Sigma_{E'}(I-A')^{-\top}
>   =\begin{pmatrix} 0.0658 & 0.0363 \\\ 0.0363 & 0.0421\end{pmatrix},
> \qquad
> \rho_{\text{int}} \approx 0.69.
> \end{align*}
> Consumption rises by \\(\sim\ 29\\%\\) and income by \\(\sim 82 \\%\\), while the
> \\(C\\)–\\(I\\) correlation falls from \\(0.75\\) to \\(0.69\\).
>
>
> > **From intervention to counterfactual via the twin SCM.**
> Above equations describe the interventional distribution $P(X_{\text{int}})$, i.e. what we would observe if the shift–scale policy were enacted for the whole population. To answer individual–level counterfactual queries (``what would this household’s consumption be had the policy applied?’’) , we follow the twin network construction:
> Let $(c,i)$ denote the actually observed consumption and income for one household.  The twin‑SCM duplicates every endogenous variable and shares the same exogenous noise:
>
> \\[
> \boxed{
> \begin{aligned}
> C &= 0.50 I + 1 + E_{C},              &  C' &= 0.50I' + 1 + E_{C},\\\
> I &= 0.40C + 0.50 + E_{I},           &  I' &= 0.8\bigl(0.40C' + 0.50 + E_{I}\bigr) + 1.
> \end{aligned}}
> \\]
>
> > where the primed copy encodes the shift–scale intervention
> $\mathrm{ss}(I,\alpha{=}0.8,\beta{=}1.0)$ and the unprimed copy remains
> factual. From the factual equations, we can solve
> \begin{equation}
> E_{C}=c-0.50i-1,
> \qquad
> E_{I}=i-0.40c-0.50.
> \end{equation}
>
> > Eliminating $C'$ from the primed equations of the twin SCM yields
> \\(
> 0.84I' = 1.72 + 0.8E_{I} + 0.32E_{C}.
> \\)
> Substituting $E_{C}$ and $E_{I}$ gives the counterfactual income \\(
> I'(c,i) = \frac{25}{21} + \frac{16}{21}i
>         = 1.190476 + 0.761905i.
> \\) Back‑substitution furnishes the counterfactual consumption
> \\(
> C'(c,i) = c + \tfrac{25}{42} - \tfrac{5}{42}i
>         = c + 0.595238 - 0.119048i. \\)
>
>
> > Above equations give the counterfactual response mapping $(c,i)\mapsto(C',I')$. Because the mapping is affine and the factual distribution $X_{\mathrm{obs}}\sim\mathcal N(\mu,\Sigma)$
> is Gaussian, the marginal counterfactual $(C',I')$ is again Gaussian with
> \\(\mathbb E[(C',I')^\top]=(2.024,2.048)^\top\\) and covariance $\begin{pmatrix} 0.0658 & 0.0363 \\\ 0.0363 & 0.0421\end{pmatrix}$.
> Here, the twin‑SCM reconciles individual counterfactual semantics
> \\((C'(c,i),I'(c,i))\\) with the population‑level interventional
> distribution already reported, while remaining in closed form due to the linearity and contractiveness of the model.
>
>
>
>
>
> ---
> (W3) & (Q2)
> > Our current framework is restricted to shift–scale interventions, but this class already strictly generalizes hard interventions and serves as a foundation for further extensions. The core theoretical principle underpinning our results is the preservation of global contractivity (i.e., the global Lipschitz constant remains below one) after intervention. This property ensures unique solvability of the intervened SCM, even in the presence of cycles, and is abstract enough to accommodate more general intervention types, provided they do not destroy contractivity. Thus, our results can be directly extended to any class of interventions (including nonlinear, stochastic, or more complex parametric changes) that preserve this contraction property. An explicit analysis of broader classes of interventions, and sufficient conditions under which they preserve contractivity, is a promising direction for future work

---

> > ### Author Response · Authors · 2025-08-01
> > **Small Correction in the Example**
> >
> > > \begin{aligned} C = 0.50,I + 1 + E_{C}, \quad I = 0.40,C + 0.50 + E_{I}, \\qquad
> > (E_{C},E_{I})^\top \sim \mathcal N(\mathbf 0, ,0.04\mathbf I_2)\end{aligned}
> >
> > should be
> >
> > > \begin{aligned} C = 0.50I + 1 + E_{C}, \quad I = 0.40C + 0.50 + E_{I},  \\qquad
> > (E_{C},E_{I})^\top \sim \mathcal N(\mathbf 0, 0.04\mathbf I_2) \end{aligned}
> >
> > **and**
> >
> > > \\[
> > \mathrm{ss}(I,\alpha,\beta):
> > I \leftarrow \alpha \bigl(0.40\,C + 0.50 + E_{I}\bigr) + \beta,
> > \quad
> > \alpha=0.8,\beta=1.0.
> > \\]
> >
> > should be
> >
> > > \\[
> > \mathrm{ss}(I,\alpha,\beta) :
> > I  \leftarrow  \alpha\bigl(0.40C + 0.50 + E_{I}\bigr) + \beta,
> > \quad
> > \alpha=0.8, \beta=1.0.
> > \\]
> >
> > We apologize for the inconvenience — some extra semicolons and commas are inadvertently included in various places.

---

> ### Author Response · Authors · 2025-08-05
>
> Thank you very much for the acceptance.

---

### Note · Authors · 2025-08-16

We thank all reviewers for their time, encouragement, and thoughtful feedback, and we appreciate their accurate summary of our work. We especially thank Reviewers vToH and Na7U for maintaining their positive acceptance, and we are grateful to Reviewer Ucnx for raising his score to acceptance. We believe we have addressed Reviewer vCep’s doubts. In the final version, we will expand our comparison with Bongers et al.—particularly their results on hard interventions—to clarify how our framework extends this foundation.
We will also include the example discussed with Reviewers vToH and vCep during the rebuttal.
Our thanks also go to the ACs and SACs for their meticulous stewardship of our submission.

---

### Decision · Program_Chairs · 2025-09-17

**Decision:**

Accept (poster)

**Comment:**

The paper develops a formalism for structural causal models that can include cycles along with a class of interventions under which variables and distributions are still well-defined, including counterfactual distributions.

Reviewers agreed the paper made a strong foundational contribution for an important problem.

The abstract should provide some simple description of shift-scale interventions.

One or more simple examples should help motivate the theoretical framework. Authors gave one in the discussion, this should be polished and included in the paper. Authors should consider whether any figures could help increase understanding for some readers.

There is some important discussion not currently in the paper that should also be added. What are the **semantics** of interventions and counterfactuals? The mathematical framework seems agnostic about whether cycles represent e.g. time dynamics or a time-independent steady state solution, so perhaps the answer to the previous question requires first making additional assumptions about which setting is under consideration. The semantics of counterfactuals in a DAG are well-known: "The value variable $x_i$ would have taken had some counterfactual condition held for other causally-related variables." Does the present framework allow any meaningful semantics, or is an intuitive/linguistic understanding precluded by e.g. infinite recursion?